# Design and Experimental Analysis of Connections for a Panelized Wood Frame Roof System

**Md Saiful Islam** [1,*] , **Ying Hei Chui** [1] and **Mohammed Sadiq Altaf** [2]

1   Department of Civil and Environmental Engineering, University of Alberta, Edmonton, AB T6G 1H9, Canada; yhc@ualberta.ca
2   ACQBUILT Inc., 4303 55 Ave NW, Edmonton, AB T6B 3S8, Canada; sadiqa@acqbuilt.com
*   Correspondence: mdsaiful@ualberta.ca

**Abstract:** This paper presents the results of an experimental study on the short-term mechanical performance of timber screw connections comprising two types of fasteners suitable for a novel panelized roof design process. Thirty-seven specimens of five different connection configurations were tested under quasi-static monotonic loading. The main objective of this study is to provide a preliminary assessment of connection capacity that is key to the successful implementation of a proposed panelized roof design method. It also provides the basis to assist in the development of a numerical model of the novel roof assembly. Additionally, the experimental data are used to check the validity of existing analytical approaches for predicting the strengths of screwed connections comprising engineered wood members. The validation exercise shows that available analytical models can be used to predict the connection capacity of the novel panelized roof.

**Keywords:** panelized roof; laminated strand lumber; home manufacturing; wall framing stations; self-tapping screw

## 1. Introduction

Panelized fabrication is a form of the off-site construction method. This process subdivides a building model into subassemblies such as wall panels and floor panels to manufacture buildings in a factory environment. Two-dimensional construction of building elements makes it more flexible in various architectural designs with a trade-off that substantial on-site workload in contrast to modular construction. In Canada, all panelized building manufacturers implement digital CAD drafting for building design and adopt automated and semi-automated manufacturing processes [1]. Typically, a light frame panelized house production facility encompasses several workstations, e.g., wall framing station (Figure 1a), sheathing assembly section, floor production line, and roof assembly section. These production lines fabricate the corresponding components using Building Information Modelling (BIM) and CNC machines, which make the process more efficient with minimal wastage of material and higher productivity. As a result, this panelized construction method is faster, and superior compared to the traditional stick-built process. In current residential house construction, the roof structure typically consists of a series of triangulated trusses fabricated with dimension lumber that is connected with light-gauge truss plates. These triangulated trusses are often fabricated in a factory and brought to the building site. The final structure is constructed by connecting the triangulated trusses using wood-based sheathing panels. Because of this type of roof system, the present light frame offsite house construction method is classified as partially panelized since the roof production process is the same as the traditional stick-framing process, with the only difference being they are built within factory space and transported to the site as a volumetric module(s) (Figure 2).

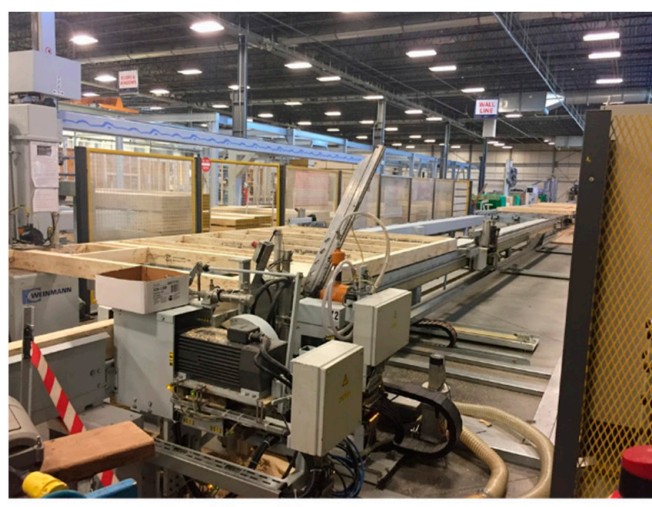 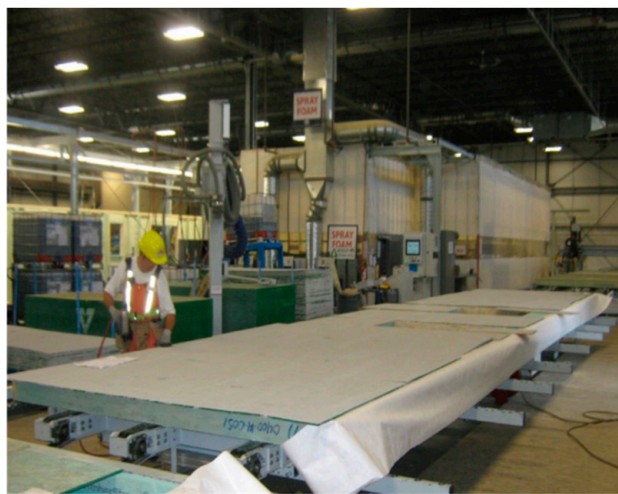

(a) Wall framing station　　　　　　　　　　　　　(b) Sheathing station

**Figure 1.** (**a**) Framing, (**b**) sheathing stations of wall production line (courtesy of ACQBUILT, Inc., Edmonton, AB, Canada ) [2].

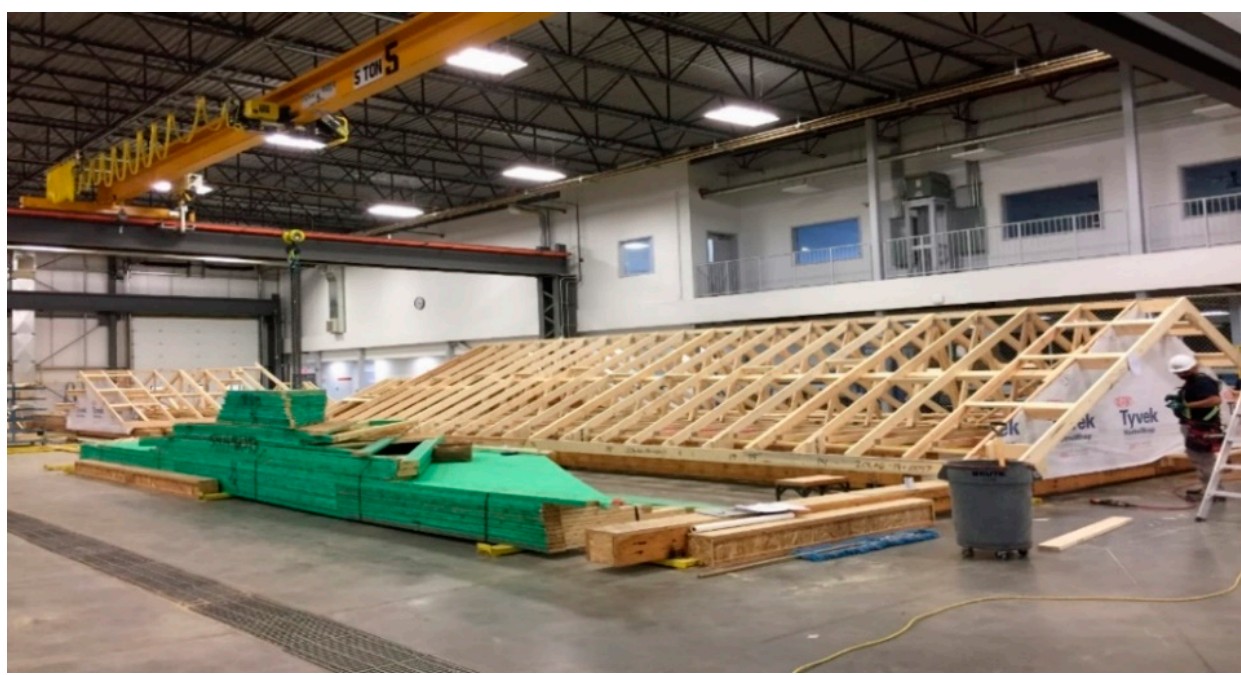

**Figure 2.** A roof production line (courtesy of ACQBUILT, Inc., Edmonton, AB, Canada) [2].

This process of roof production is significantly less productive in contrast to the other component production stations such as wall and floor lines. In fact, from the manufacturing perspective, roof workstation imbalances the overall production rate, taking into account the effect on fabrication efficiency at other workstations. For instance, an Alberta-based home manufacturer's wall production station is capable of producing wall panels for three homes in an 8 hour shift, whereas one roof-framing workstation produces only one complete roof [3]. A time study of current roof manufacturing stations shows that one complete gable roof production requires an average of 82.5 man-hours in contrast to only 6 man-hours for the walls [4].

The penalization of roofs is an improvement to current factory-based housing construction that can be introduced to further reduce the cost of houses. The prefabricated roof panels need to be connected on-site to make this innovative panelized roof system

perform structurally. Therefore, the development of various panel connections is a key part of the research leading to the implementation of the panelized roof systems. A study was conducted to develop these connection details, which is the subject of this paper. Gable roofs are arguably the most common roof structure for single-family residences, garages, barns, warehouses, and factories in North America. Hence, this study focuses on the connection development of the new panelized-roof system for the gable-type roof that takes into account manufacturing factors and on-site assembly efficiency.

In the development of the required connection details, the entire gable roof was divided into several subsections. The dimensions of these subsections must comply with the production line constraints, transportation trailer capacity, crane lifting limitations, and on-site installation considerations. The resulting system for a typical two-storey house with a gable roof comprises the following components (Figure 3): (a) roof panels, (b) support wall panels, (c) ceiling frames, (d) beams spanning over two support walls, (e) gable ends, and (f) inter-component connections, including the inclined roof panel-to-support wall, ceiling frame-to-load-bearing shear wall, and the support wall-to-ceiling frame.

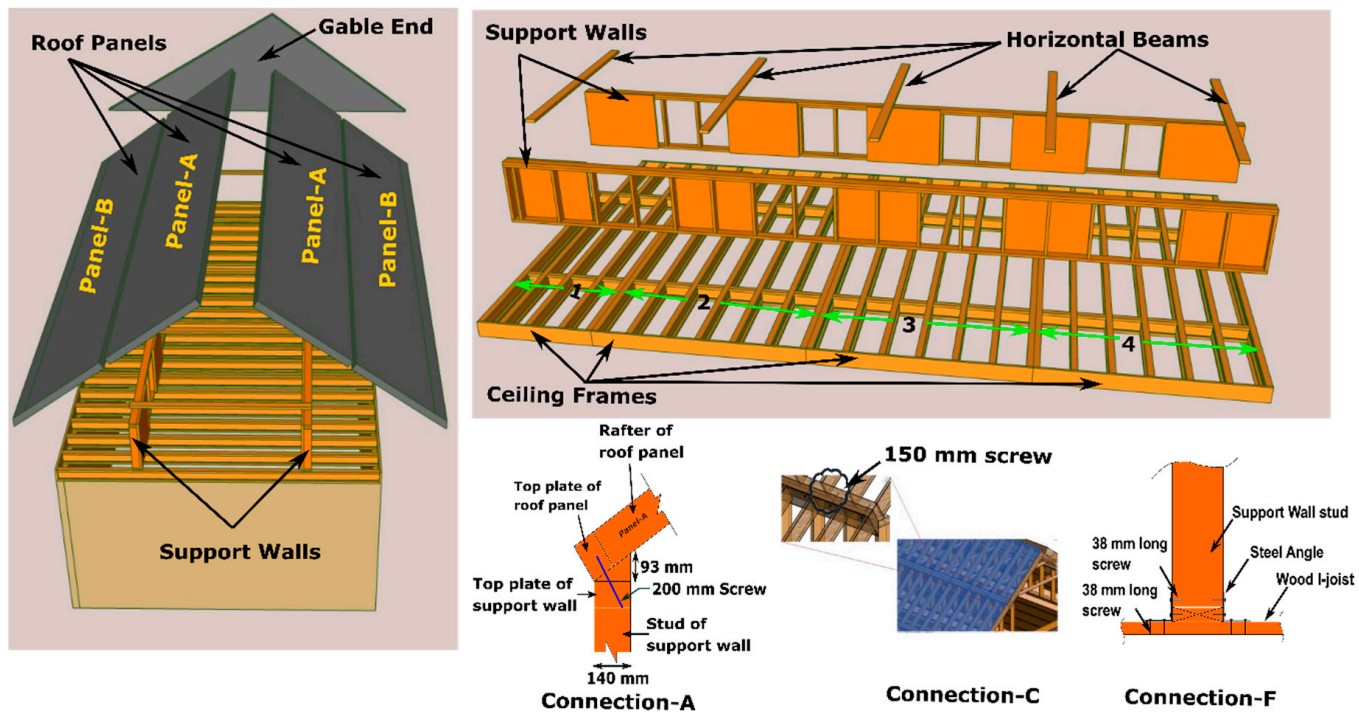

**Figure 3.** Panelized roof concept.

The roof panels and support walls are produced in the wall production line using laminated strand lumber (LSL) and Oriented Strand Board (OSB). Whereas the ceiling frames are produced in the floor line using wood I-joists. Details of the panel and ceiling frame design are discussed in Islam et al. [5,6]. In brief, the production process of the roof panels is the same as the wall production; for example, the framing operation is performed in the framing station (Figure 1a) and attaching the OSB to the finished frame is completed in the multifunction bridge (Figure 1b) of the wall line. Other components such as roof panel to support wall connection requires a combination of automated and manual operations. For instance, wedge pieces for connection can be produced on a CNC-controlled processing machine which is capable of cutting at any angle and installing screws manually.

## 2. Development of Connection Details

The complete panelized roof is a complex three-dimensional (3D) system, that consists of an assembly of several components, where inter-component connections are a significant factor in the effective implementation of this design. In this system, all the

panels are assembled at the site, and the inter-component connections must be easy to install with a minimum workload. In total, the panelized roof has the following eight primary joint types (Figure 4): (i) Connection-A: Support wall-to-panel-A connection; (ii) Connection-B: Support wall-to-wedge connection for panel-B; (iii) Connection-C: Panel-to-panel connection; (iv) Connection-D: Panel-B-to-ceiling frame connection at the eave line; (v) Connection-E: Apex line connection; (vi) Connection-F: Support wall-to-ceiling frame connection; (vii) Connection-G: Gable end-to-roof panel, (viii) Connection-H: Gable end-to-support wall.

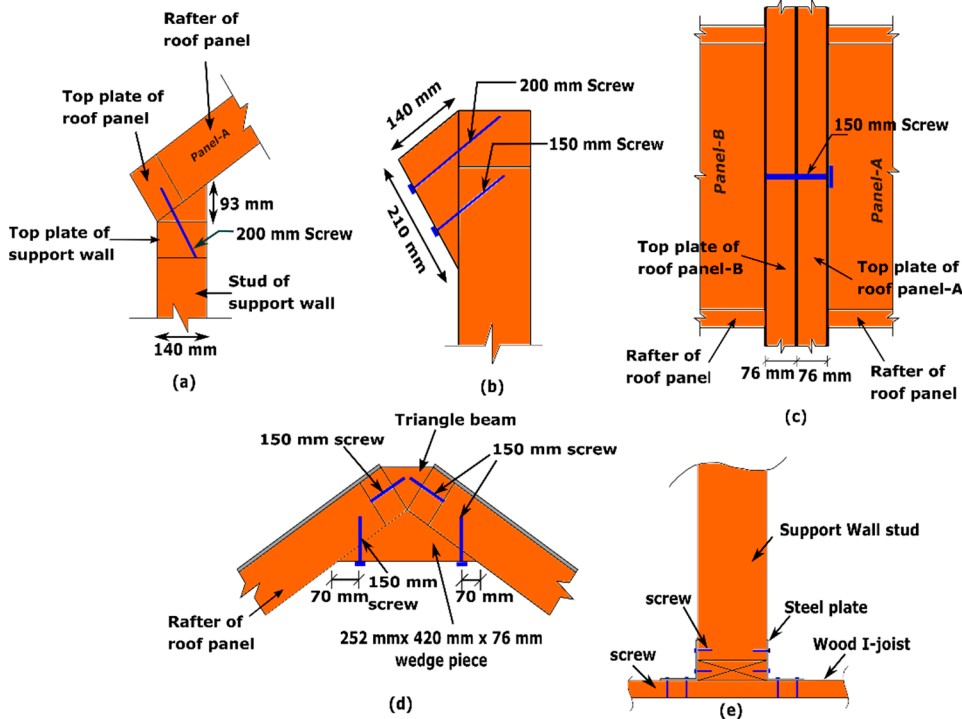

**Figure 4.** (**a**) Connection-A and Connection-D, (**b**) Connection-B (**c**) Connection-C, (**d**) Connection-E, (**e**) Connection-F.

Connection-A comprises a 200 mm screw connecting the top plates of the support wall and roof panel (Figure 4a), whereas panel-B is supported on a wedge piece (210 mm × 140 mm × 76 mm) attached to the support wall using two screws (200 mm and 150 mm long) in the case of Connection-B (Figure 4b). Connection-C is fabricated using a 150 mm screw to connect the top plate of the roof panels laterally (Figure 4c). A similar joint configuration as Connection-A can be used for panel-B-to-ceiling frame connection (Connection-D) at the eave line. Connection-E consists of a 12 ga 90° steel angle plate and 38 mm long screws (Figure 4e). These screws are commercially available and an alternative to common 10d nails with a higher load capacity. The apex connection consists of a 150 mm screw joining the two opposite side framing rafters of the roof panel to a 252 mm × 420 mm × 76 mm wedge piece, and the top plate of the two roof panels is connected to a triangular-shaped beam via 150 mm screws (Figure 4d). Finally, the gable end-to-roof panel connection is a timber-to-timber screw connection with a 150 mm long screw, whereas in the case of the gable end-to-support wall connection, screws similar to the Connection-E are utilized (Figure 5).

The location and corresponding load case of the connections explained previously are shown in Figure 6. Depending on the load cases the connections are subjected to lateral or withdrawal load. For instance, Connection-A and D are subjected to lateral loading along the inclined plane of the wedge in the gravity load case (Figure 7a) whereas withdrawal in the wind uplift load case. For Connection-B, it is required to determine the shear capacity along the vertical plane of the support wall (Figure 7c), while connection-C is subjected to lateral load in two different strand directions of LSL (Figure 7d,e). Connection-E is

subjected to shear loading in the gravity load case but a combined axial and lateral load in the wind load case (Figure 7g). Connection-F resists shear load for the angle plate side attached to the support wall and screws withdrawal force for other right-angle parts of the connection, as shown in Figure 7h. Therefore, it is evident that experimental evaluation is required to generate test data that can be used to assess their load-carrying capacities for structural design purposes. An experimental study was conducted to test all connections except Connection-G and Connection-H as they are similar to laterally loaded timber-to-timber connections and capacity assessment can be performed using the analytical model available in timber design standards, such as CSA O86-19 [7].

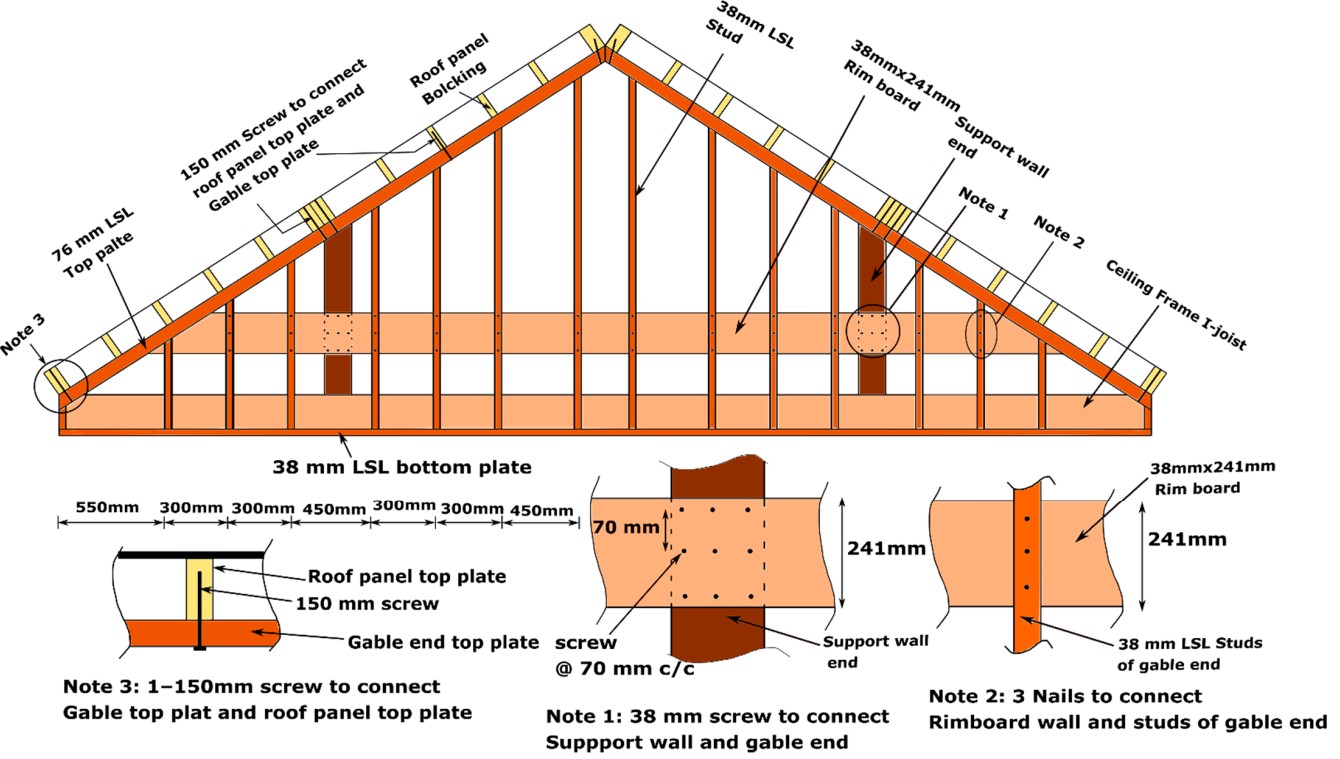

**Figure 5.** Connection-G and H.

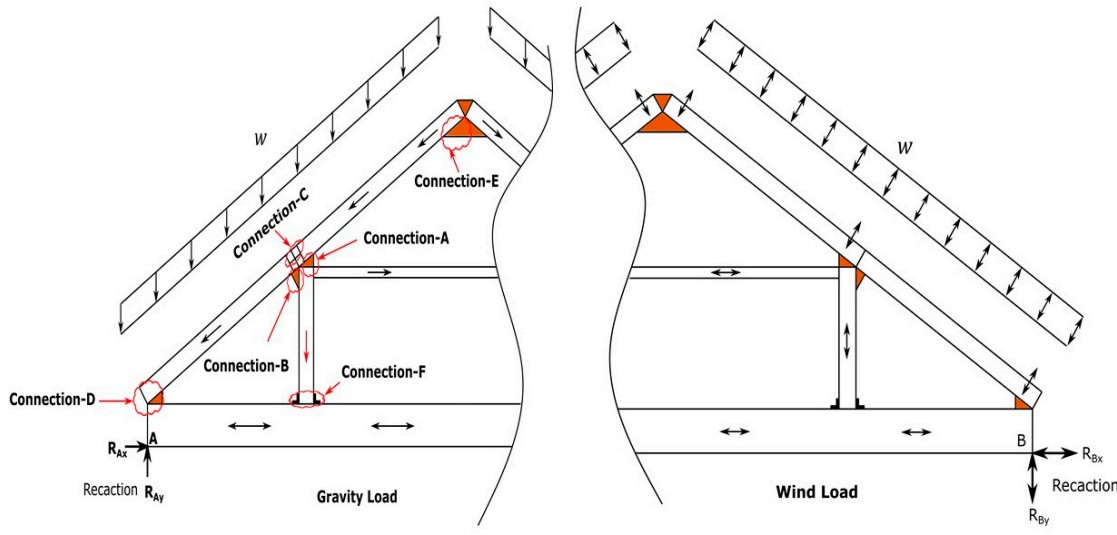

**Figure 6.** Connection location and load cases.

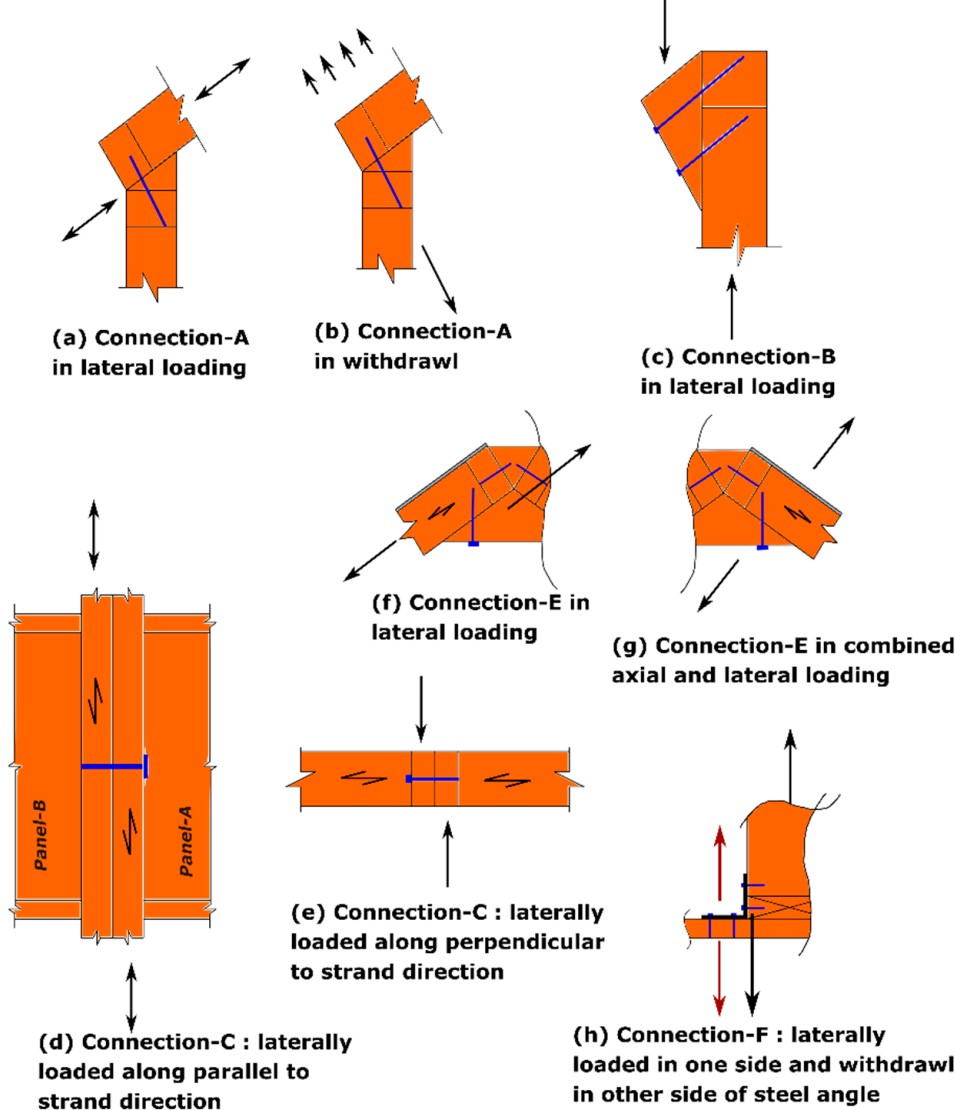

**Figure 7.** Loading conditions of the connections.

## 3. Materials

### 3.1. Screws

There were two types of screws employed in the experimental study: (a) partially threaded self-tapping screw primarily used in mass timber products (Rothoblaas HBS) and (b) timber screw with flat head predominantly used in light-frame wood construction as an alternative to bolts and traditional lag screw (Simpson Strong-Tie SDWS) for Connection-A and Connection-B [8,9]. The geometries of the HBS and SDWS screws are quite similar to each other, with a milling cutter between the thread and the shank and a pronounced cutter on the tip. The main difference between the two screws lies in the shape of the head, with a countersunk head in the case of the HBS screw. The dimensions and the mechanical properties provided by the relevant European Technical Approval (ETA) [10] and Uniform Evaluation Service [11] reports are summarized in Table 1. For Connection-C, only the SDWS screw was used in this experimental study since the screw manufacturer's [12] technical data does not have the shear capacity for engineered wood products. The Connection-E test specimen was fabricated using HBS screws only since it can be driven faster than SWDS screws. It is worth mentioning that half of this connection installation will be performed at the site, and faster screw installation is preferred.

**Table 1.** Screw geometry and properties.

| Connector | HBS Screw | | SDWS Screw | |
|---|---|---|---|---|
| Commercial name | HBS6150 | HBS6200 | SDWS22600DB | SDWS22800DB |
| Fastener length (mm) | 150 | 200 | 152 | 203 |
| Shank length (mm) | 75 | 125 | 82 | 133 |
| Head diameter (mm) | 12.0 | 12.0 | 19 | 19 |
| Shank diameter(mm) | 4.3 | 4.3 | 5.5 | 5.5 |
| Nominal diameter (mm) | 6.0 | 6.0 | 7.7 | 7.7 |
| Tip diameter (mm) | 3.95 | 3.95 | 5.0 | 5.0 |
| Head thickness (mm) | 4.5 | 4.5 | - | - |
| Characteristic yield moment (N-mm) | 9494 | 9494 | 25,590 | 25,590 |
| Characteristic tensile strength (N) | 11,300 | 11,300 | - | - |
| Allowable tensile strength (N) | - | - | 7006 | 7006 |

### 3.2. Timber Elements

As previously mentioned, the framing members of the roof panels are structural composite lumber and dimension lumber. The structural composite lumber was Laminated Strand Lumber (LSL) of 1.30E grade [13]. The product is manufactured from wood strands blended with an isocyanate-based binder adhesive. The manufacturing process of LSL primarily orients the wood strands parallel to the long axis of the mat within a range of ±10° [14]. As a result of cross laminating between cells in wood strands, LSL has high fracture toughness relative to most other solid wood products [15]. The mechanical properties and density of the LSL from product documentation [16] and experimental data are reported in Table 2.

**Table 2.** LSL properties.

| Grade | 1.30 E |
|---|---|
| Modulus of Elasticity (MOE) (MPa) | 8965 |
| Density (kg/m$^3$) | ≥624.72 |
| Density (experimental) (kg/m$^3$) | 698.80 |
| Equivalent Specific gravity (connection design), SG | 0.5 |

## 4. Fabrication of Connection Specimens

The complete assembly of the Connection-A specimen is comprised of three components, such as 140 mm × 205 mm × 76 mm (representative element of roof panel -A), 140 mm × 330 mm × 76 mm (representative element of support wall top plate) LSL cut pieces, and 93 mm × 140 mm × 76 mm LSL wedge pieces (Figure 8). The wedge piece was first glued on the top pate element and a 200 mm screw was inserted at an angle of 30° with the vertical direction of the stud to connect the roof panel part (Figure 8). A total of 23 LSL pieces were fabricated to obtain seven representative samples for the shear capacity test and nine samples for the wind uplift test of this connection type.

Connection-B controls the slope of the roof panel, and it had the most complex fabrication setup. It had two components, i.e., a small representative element of the support wall panel and the wedge piece. First, a total of 16 225 mm × 451 mm small panels (Figure 9a) were fabricated using two 38 mm LSL studs, a 76 mm top plate, and 9.5 mm OSB that represent a sample of the support wall, and then the wedge piece was connected using two self-tapping screws (STS) as shown in Figure 9b, which resembles the actual connection.

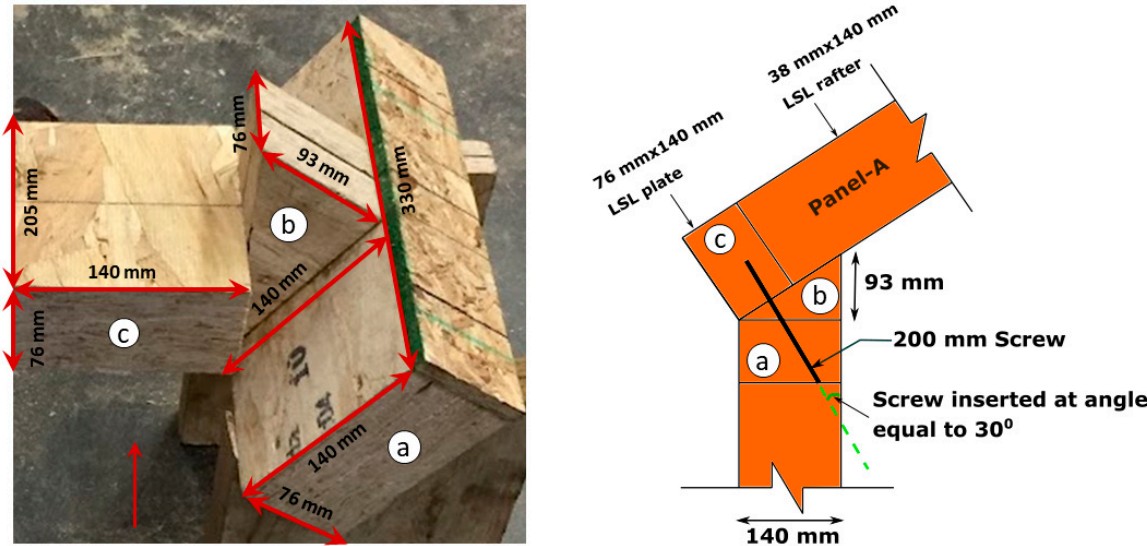

(a) Rrepresentative element of Support wall top plate stud　(b) Wedge piece　(c) Rrepresentative element of roof panel -A

**Figure 8.** Connection-A specimen.

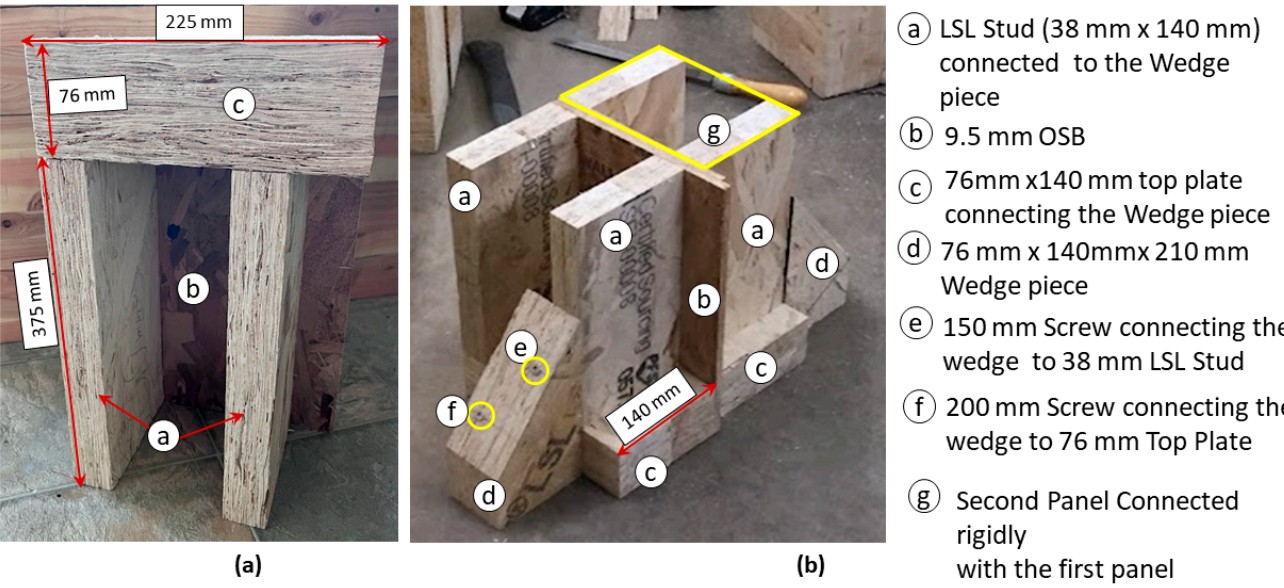

(a) LSL Stud (38 mm x 140 mm) connected to the Wedge piece

(b) 9.5 mm OSB

(c) 76mm x140 mm top plate connecting the Wedge piece

(d) 76 mm x 140mmx 210 mm Wedge piece

(e) 150 mm Screw connecting the wedge to 38 mm LSL Stud

(f) 200 mm Screw connecting the wedge to 76 mm Top Plate

(g) Second Panel Connected rigidly with the first panel

**Figure 9.** (**a**) 225 mm × 410 mm panel fabrication (**b**) complete Connection-B specimen.

　　　To create a symmetric pushout test setup, two panels were connected side by side rigidly using small screws so that the assembly acts as a single piece (Figure 9b), and thus, the fabricated sample ensured the designed test setup. Additionally, six panels (225 mm × 451 mm) were also fabricated for Connection-E specimens (Figure 10). The difference, in this case, was the use of a 150 mm long screw to connect the wedge and the roof panel rafter (Figure 10). It is worthwhile to note that the other part of the apex connection with the triangular beam shear test is similar to the panel-to-panel connection.

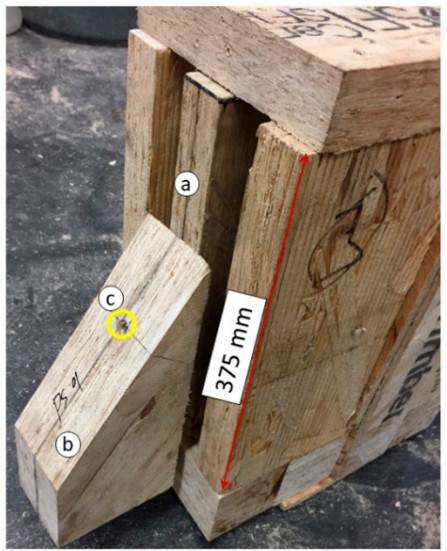

**Figure 10.** Connection-E specimen.

Connection-C samples were fabricated by connecting two pieces of 76 mm × 140 mm LSL side by side using SWDS timber screw in two different orientations as explained in the test setup section. Connection-F specimens were fabricated by attaching a 38 mm × 140 mm × 600 mm LSL stud with two 76 mm × 140 mm × 200 mm LSL blocks using 12-gauge angle plates and screws as illustrated in Figure 11.

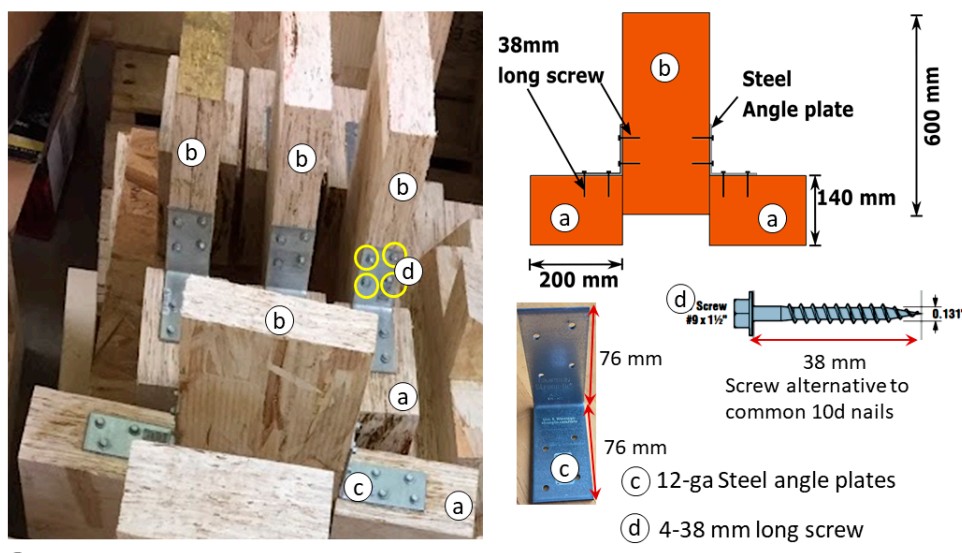

**Figure 11.** Connection-F specimen.

## 5. Test Setup

ASTM D1761-12 [17] provides a guideline for the timber screw connection test. According to this standard, the test specimen was subjected to quasi-static monotonic loading at a rate of 2.54 mm/min. Because of the specific orientation of the connection specimens, modified test setups were prepared for each connection type. Although ASTM D1761-12 [17] states that displacement should be recorded up to the first drop of load and a slip limit of 15 mm is recommended by EN 26891 [18] for ultimate condition, where possible, the specimens were loaded to their actual failure limit state to evaluate the residual capacity. Specimens were tested using an MTS Test Machine with a 1000 kN capacity under displacement control. The proceeding section discusses individual test setup.

### 5.1. Connection-A Setup

As can be observed in Figure 7a, Connection-A is a single shear connection. Determination of shear capacity requires loading parallel to the plane of the wedge element. Therefore, two connection specimens were placed side by side (Figure 12a) on the bottom base steel plate as shown in Figure 12b. Then the representative elements of roof panel-A were attached by threaded steel rods passing through the top and bottom steel plates (Figure 12b). The representative elements of the support wall top plate were clamped rigidly at the bottom by four threaded steel rods passing through the base steel plate, and an L shaped angle bracket as illustrated in Figure 12b. Two cable transducers measured the slip between the central part and the side members.

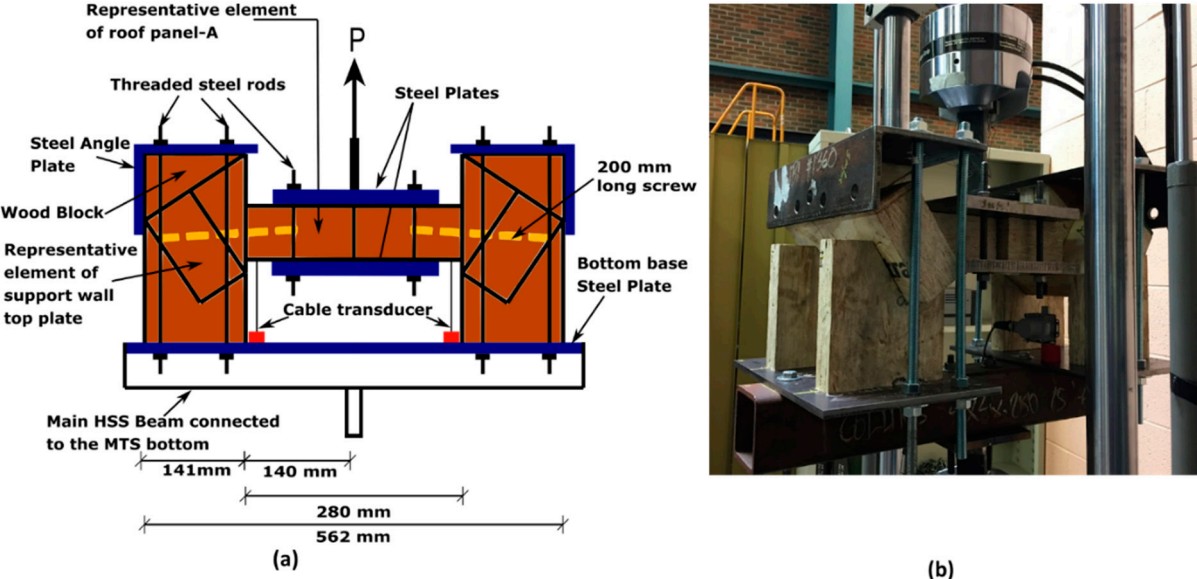

**Figure 12.** Connection-A test setup for shear loading: (**a**) schematics; (**b**) actual image.

The screw withdrawal test apparatus is shown in Figure 13. In this case, only one test specimen was placed on the base steel plate (Figure 13). The representative element of the support wall top plate part was attached to the base plate using threaded steel rods (Figure 13). While the representative element of the panel-A part was connected to the crosshead of the MTS machine by the same mechanism used for the bottom part. MTS head displacement was used to calculate the slip between the connection members.

### 5.2. Connection-B and Connection-E Setup

Connection-B is also a single-shear connection with two screws. The test apparatus to determine the capacity was a modified typical timber connection push-out setup. As can be observed from Figure 14, the wedge parts were supported on a small HSS beam, and the L shaped angle plate was clamped to avoid any lateral movement. A steel plate was used to distribute the load evenly on the top of the specimen. Two cable transducers were placed on both sides of the specimen to record the relative slip between the central part and the side members. Connection-F specimens were also tested for shear loading using the same apparatus. Two cable transducers measured the slip of the middle member relative to the side members.

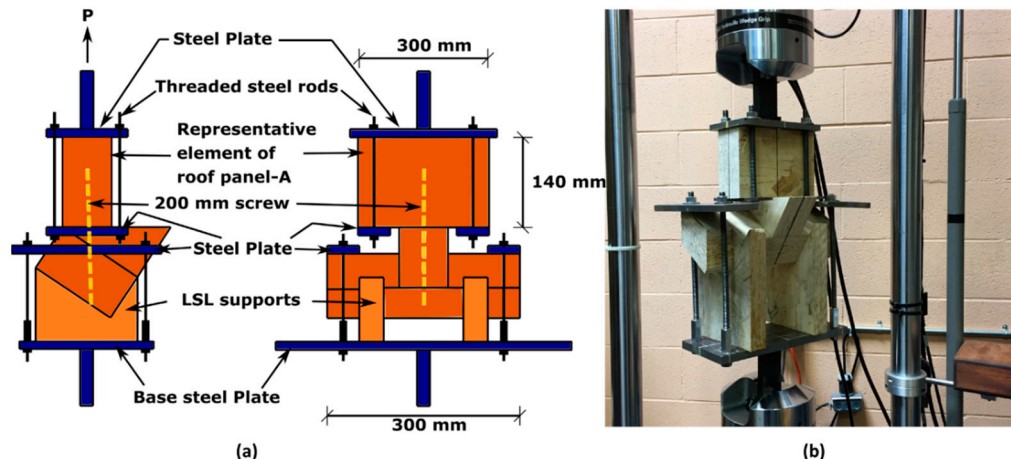

**Figure 13.** Connection-A withdrawal test setup: (**a**) schematics; (**b**) actual image.

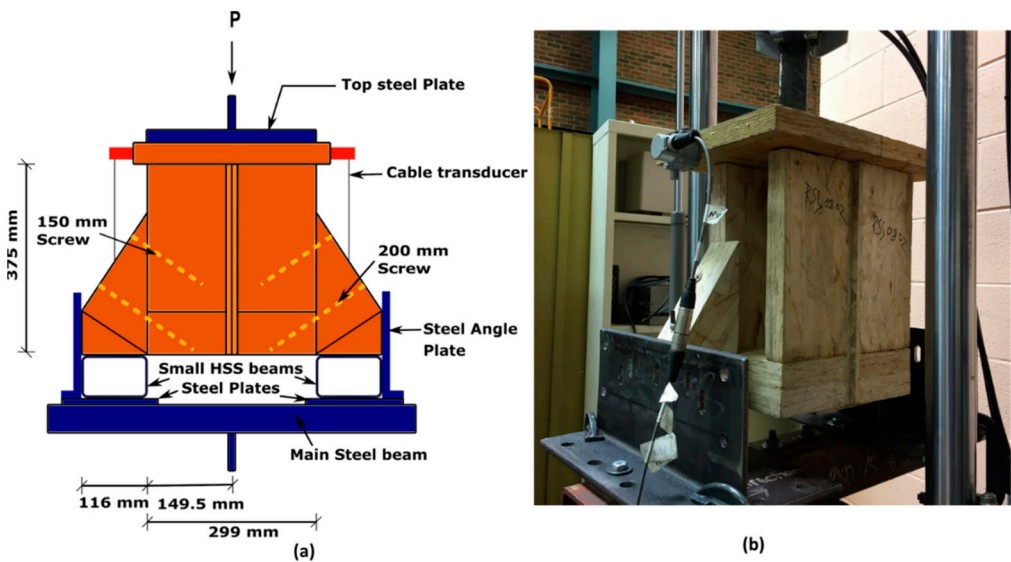

**Figure 14.** Connection-B test setup: (**a**) schematics; (**b**) actual image.

### 5.3. Connection-C Setup

The Connection-C test assembly was typically pushed out in a single shear plane but two different orientations of the specimen, i.e., (a) loading parallel to the LSL strand direction and (b) loading perpendicular to the LSL strand direction. To reduce minimize friction between the moving side LSL member and the steel beam face, a polypropylene sheet was placed as indicated in Figure 15. Two LVDTs measured the relative slip between the connection members.

### 5.4. Connection-F Setup

For this type of connection, the screw manufacturer guide provides shear capacity where screws are inserted on the wider face of timber studs. However, for Connection-F it is required to insert the screws on the narrow face of the LSL. The withdrawal capacity of this connection is available in the screw manufacturer's technical data report [19]. Therefore, a test setup was developed to determine only the shear capacity of the connection. The side LSL parts were clamped by a threaded steel rod passing through both the bottom base steel plate and upper steel plate as shown in Figure 16, whereas the middle LSL part was connected to the crosshead of MTS by the same mechanism. Two LVDTs were placed on either side of the specimen to record the relative slip between the central and side LSL members.

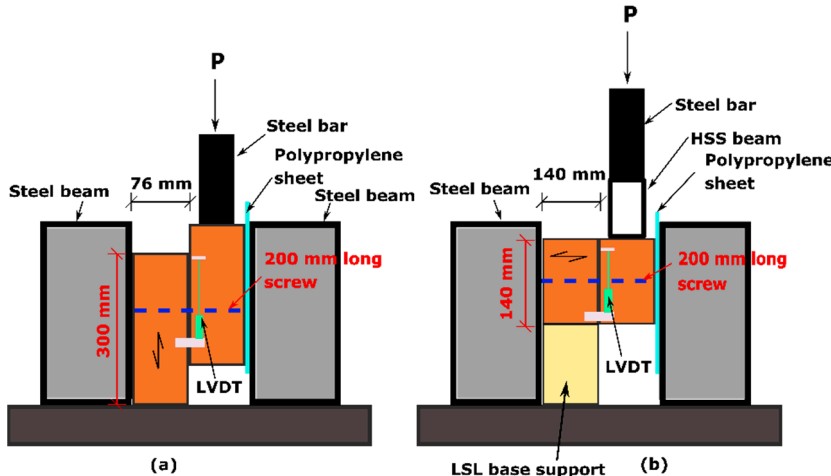

**Figure 15.** Schematics of Connection-C test setup: (**a**) loading parallel to strand; (**b**) perpendicular to strand.

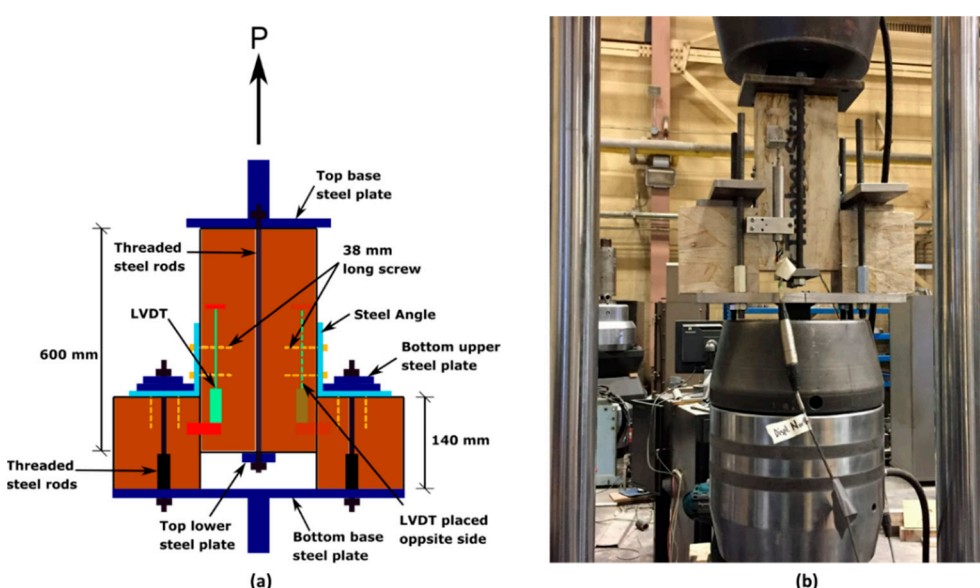

**Figure 16.** Connection-F test setup: (**a**) schematics; (**b**) actual image.

## 6. Assessment of Connection Mechanical Parameters

The connection strength in this study was determined according to EN 26891 [18], which defines it as the maximum load before failure of the specimen if the corresponding slip value is less than 15 mm or, otherwise, the load value when slip value is 15 mm. The evaluation of the slip modulus, $K_s$, of all the connections (corresponding to the slip modulus $K_{ser}$ provided by EN 1995-1-1 [20]) was calculated by means of the following equation:

$$K_s = \frac{0.4F'_{max} - 0.1F'_{max}}{v_{0.4} - v_{0.1}} \tag{1}$$

where $v_{0.4}$ and $v_{0.1}$ are the connection slips corresponding to loading equal to $0.4F'_{max}$ and $0.1F'_{max}$, respectively; $F'_{max}$ is the mean value of the peak load values, $F'_{max,i}$ recorded peak load for all test repetitions for each connection type [21–23]. The yield load of a timber connection ($F_y$) is the load value corresponding to the entry into the plastic field. Identifying yield load requires a clear demarcation point between the linear-elastic and plastic regions of the load-slip curve. However, for all load-slip curves, the demarcation point was not easy to identify. Therefore, the ASTM D5764 5% offset method [24] was used to determine the

yield point. The 5% offset method defines the yield point as the intersection of a straight line parallel to the initial linear line part (slope between 0% and 40 % of the peak load). The straight parallel line is drawn at an offset of 5% of the diameter of the fastener.

## 7. Results

All the connection test specimens exhibited peak load capacity, and then, the loads gradually decreased while deformations increased, except Connection-A specimen fabricated using HBS screws; none of the test specimens failed suddenly with a rapid loss of applied load. The connection performance parameters, such as peak load and slip modulus that were derived from the test data are reported in Table 3. The test mean maximums reported are normalized by the whole connection, not per screw. For every parameter, the coefficient of variation (*CoV*), is given. The load-slip curves were obtained by taking the average displacement of two cable transducers/LVDT and the applied load to the specimen which was recorded by a load cell. After the connection test, selected specimens from all the types of connections were cut and opened to examine the deformed shape of the screw, which revealed the failure mode of the joint. In addition to the maximum shear load of the connection specimens, the yield load for each test was also calculated to determine the load at which the connections begin to deform inelastically.

**Table 3.** Test results.

| | $F_{max}$ per Connection | | $K_s$ (N/mm) | $F_y$ (kN) |
|---|---|---|---|---|
| | *Mean* (kN) | *CoV* (%) | | |
| Connection-A in shear loading | | | | |
| HBS screw | 4.53 | 15.60 | 823 | 3.17 |
| SDWS Screw | 5.70 | 13.31 | 1417 | 3.37 |
| Connection-B in shear loading | | | | |
| HBS screw | 12.84 | 10.8 | 3460 | 5.89 |
| SDWS Screw | 21.27 | 14.90 | 5834 | 12.42 |
| Withdrawal test of Connection-A | | | | |
| HBS screw Withdrawal | 5.88 | 17 | 4401 | 4.38 |
| SDWS Screw Withdrawal | 9.20 | 8.43 | 5136 | 5.61 |
| Connection-C | | | | |
| Loading parallel to the LSL strand | 13.72 | 21.00 | 5566 | 6.54 |
| Loading perpendicular to the LSL strand | 11.70 | 20.94 | 5596 | 6.5 |
| Connection-E | | | | |
| Screw and steel side plate | 5.46 | 3.72 | 3231 | 2.51 |
| Connection-F | | | | |
| HBS screw | 5.77 | 17.59 | 3310 | 3.76 |

### 7.1. Connection-A Results

Figure 17 shows the load-slip curve of the Connection-A test in shear loading. The connection strength and stiffness of the SDWS screw are higher than those of the HBS screw. However, assessment of the failed specimen revealed that the SDWS screw broke inside the point side member, which explains the sharp drop in load after peak value. As can be observed from Figure 18, the failure mode of the HBS screw was yielding followed by two plastic hinges per shear plane, and the bending of the screw continued even after 35 mm of deformation. Thus, the maximum connection strength of the HBS screw connection was defined at 15 mm slip. Figure 19 illustrates the load-slip curve of Connection-A in

withdrawal. The failure mode of the HBS screw was head pull-through in the side member. As a consequence, there was no well-defined peak point.

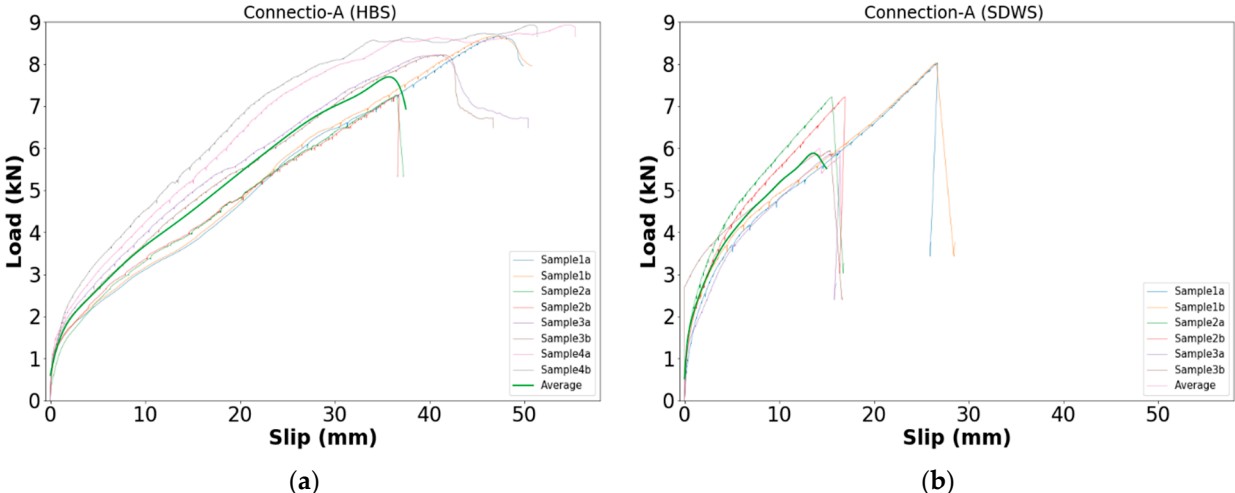

**Figure 17.** Load-slip curve of Connection-A: (**a**) HBS screw; (**b**) SDWS screw.

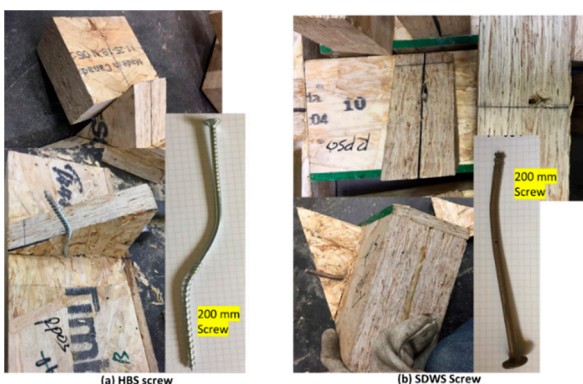

**Figure 18.** Failure of mode of Connection-A in shear loading: (**a**) HBS screw; (**b**) SDWS screw.

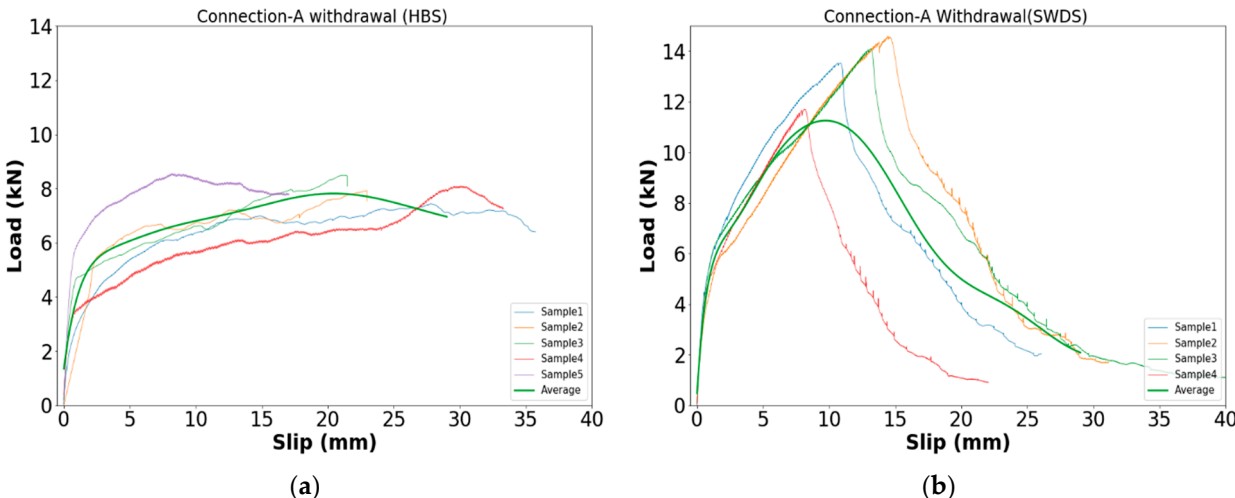

**Figure 19.** Load-slip curve of Connection-A in withdrawal: (**a**) HBS screw; (**b**) SDWS screw.

Although ASTM D1761-12 [17] recommends withdrawal resistance at the peak load value of the test, here maximum withdrawal capacity was defined at 5 mm slip since for standard withdrawal tests of these types of screws have been reported by Li et al. [21],

Gutknecht and Macdougall [25], and Xu et al. [26] at slip values of 4 to 6 mm. For SDWS screw Connection-A, specimen failure mode was the withdrawal of the threaded part of the screw.

### 7.2. Connection-B Results

For all Connection-B specimens, a crack in the LSL side member was observed after reaching the peak load. Not surprisingly, the connection specimens fabricated using SWDS screws had a higher peak load value than those made with HBS screws, as they have larger nominal diameter and fastener yield moment capacity (Figure 20). However, the HBS screw connections exhibited more ductile behaviour in contrast to the SDWS screw. Specimen cut of SDWS screw connection revealed the shear failure of the screw itself at the transition location of the beginning of the thread and smooth shank in the case of all Connection-B specimens. As can be observed in Figures 21a and 22, two plastic hinges were formed on both HBS screws, while for the 150 mm long SDWS screw, one plastic hinge was detected. In addition to crack development, head pull-through of the fastener into the side member (wedge piece) was also observed in the case of Connection-B made with HBS screws (Figure 21a). Whereas in the case of the SWDS screw, only crack formation in the side member (wedge piece) and shear failure of the 200 mm long screw inside the main member were noticed.

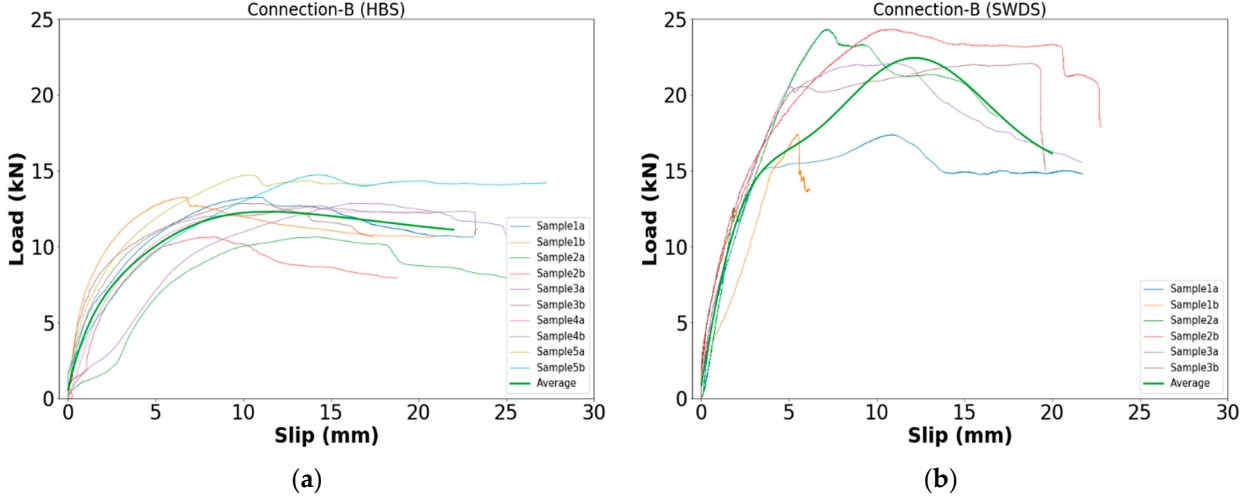

**Figure 20.** The load-slip curve of Connection-B: (**a**) HBS screw; (**b**) SDWS screw.

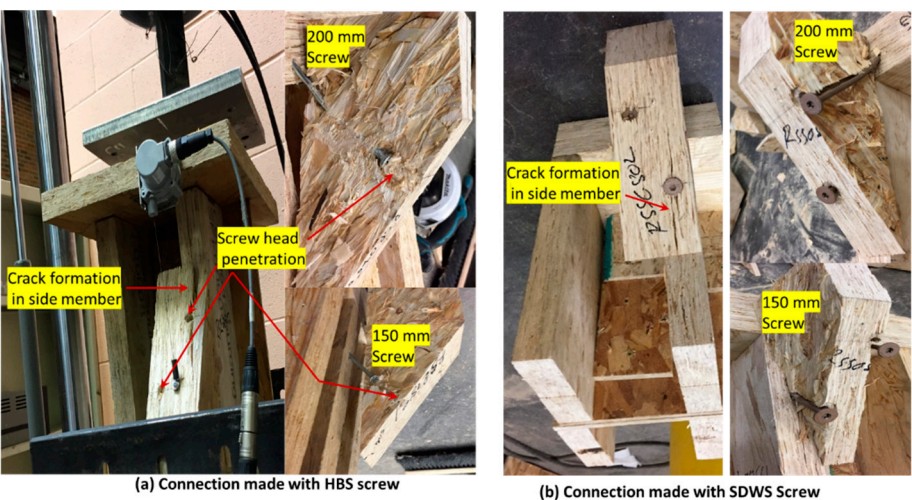

**Figure 21.** Failure of Connection-B: (**a**) HBS screw; (**b**) SDWS screw.

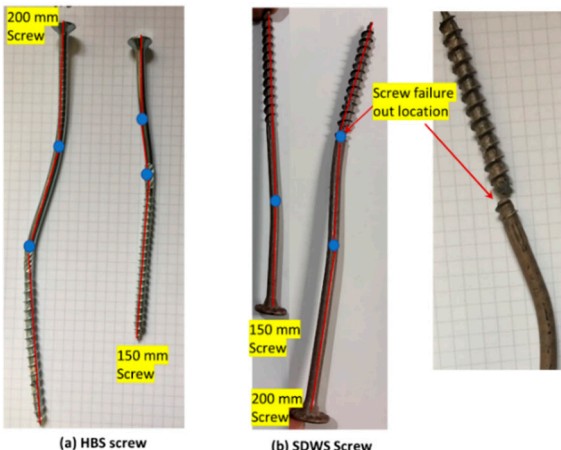

**Figure 22.** Failure mode: plastic hinge formation of the screws in Connection-B: (**a**) HBS screw; (**b**) SDWS screw.

### 7.3. Connection-C Results

Figure 23 shows the load-slip curve of Connection-C specimens. There was an insignificant difference in the peak load values for both loading directions. This may be due to the cross-laminating structure of wood strands in LSL, which distinguishes it from natural timber. The failure mode of this connection was the yielding of the SDWS screw and ultimately the breakout of the screw inside the point side member (Figure 24).

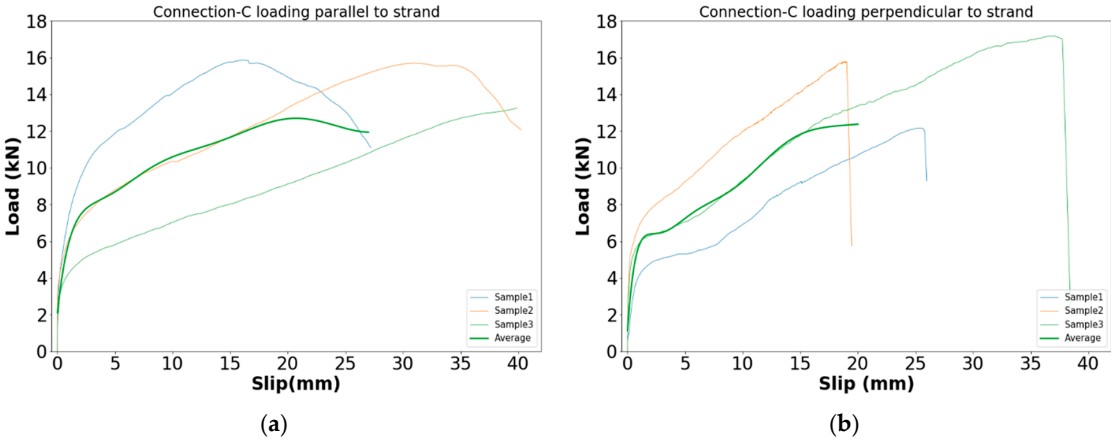

**Figure 23.** Load-slip curve of Connection-C: (**a**) loading perpendicular to the strand; (**b**) loading parallel to the strand.

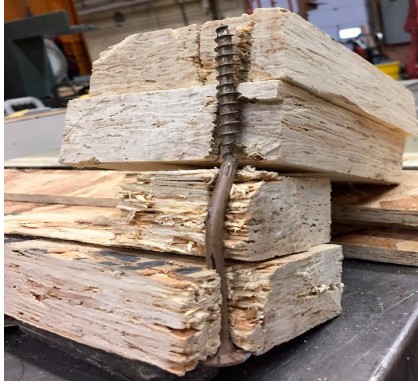

**Figure 24.** Failure mode of Connection-C.

### 7.4. Connection-E Results

As can be observed from Figure 25, all the Connection-E specimens exhibited peak load with an average value of 5.77 kN. The failure mode of Connection-E was the yielding of the screw followed by head pull-through, ultimately forming two plastic hinges. No additional test was performed to assess the connection capacity in withdrawal for Connection-E as it can be estimated using the Eurocode 5 interaction equation [20]. It should be noted that in wind load, the screw is loaded in a combination of the axial and lateral directions. It was also observed in the Connection-A withdrawal test that the failure mode for HBS screw head was head pull-through. Thus, for Connection-E, it is obvious that screw axial load capacity will be governed by head pull-through. Therefore, only the lateral capacity test for Connection-E was performed.

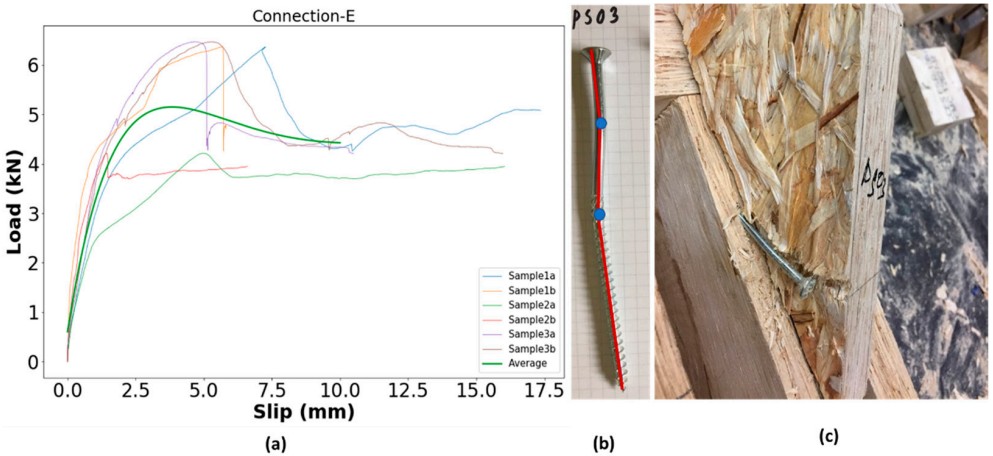

**Figure 25.** (**a**) Load-slip curve; (**b**) screw from cut specimen; (**c**) failure mode of Connection-E.

### 7.5. Connection-F Results

Connection-F is the steel-to-timber connection, and the objective of this test is to check the shear capacity of this connection as screws were inserted on the narrow face of the LSL stud. The maximum mean shear capacity obtained from the test was 5.46 kN, whereas the manufacturer's technical specification [19] reported allowable shear resistance of 3.69 kN for timber with SG of 0.5. Since mean test value is higher than the reported value, it is safe to use the design value in the specifier's guide. However, the withdrawal resistance reported by the manufacturer is 3.1 kN. Therefore, it can be concluded that in the case of Connection-F, withdrawal capacity governs. The failure mode of Connection-F was embedment failure in wood, leading to the withdrawal of the screws from the member (Figure 26).

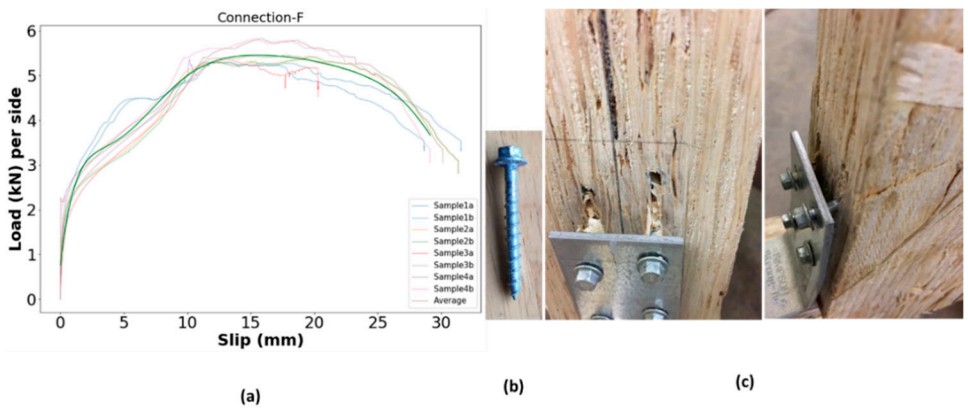

**Figure 26.** (**a**) Load slip response; (**b**) screw from cut specimen; (**c**) failure mode of Connection-F.

The data sets from this study will allow further investigation of the applicability of these connection assemblies with self-tapping screws in designing the panelized roof. As can be observed from the geometry of Connection-A and Connection-B, the angle of the wedge piece depends on the slope of the roof. This study only investigated the connection specimens for one roof slope (8/12); therefore, validation of connection capacity using analytical models is discussed in the following section. If the validation with the analytical model can predict the reasonable connection capacity, then it is obvious that for another roof slope, the developed connections can be utilized.

## 8. Comparison between Experimental Results and Theoretical Models

To facilitate connection design for any roof slope, it is essential to have a methodology that can be used by designers to predict the connection capacity. Hence, test data in terms of connection capacity are compared to the values predicted by means of theoretical models available in the literature. The load-bearing capacity ($F_{max}$) of screws inserted at 90° with respect to the shear plane can be calculated by using the theoretical model included in the CSA086 [7] or CEN [20], which is based on Johansen theory [27]. However, in Connection-B, the screws were installed at an angle of roof slope with the line perpendicular to the shear plane. A theoretical model for the estimation of the connection capacity of fasteners inserted at an angle with the shear plane was proposed by Bejttka and Blaß [28]. According to their study, the load-carrying capacity of the screws inserted at an angle $\alpha$ with respect to the line perpendicular to the shear plane consists of two components, the bearing resistance and withdrawal resistance. Each of the expressions shown below contains these two component contributions and is associated with a failure mode, with the connection capacity governed by the lowest value of the failure mode a to f:

$$F_{v,R} = min\left\{ R_a;\ R_b;\ R_c;\ R_d;\ R_e;\ R_f \right\} \tag{2}$$

$$R_{ax} = min\left\{ \begin{matrix} R_{ax,1} \\ R_{ax,2} \end{matrix} \right\} \tag{3}$$

The corresponding failure modes are:

$$(a)\ R_a = R_{ax}\ sin\alpha + f_{h,1}\ s_1 d\ cos\alpha$$

$$(b)\ R_b = R_{ax}\ sin\alpha + f_{h,2}\ s_2 d\ cos\alpha$$

$$(c)\ R_c = R_{ax}\ (\mu\ cos\alpha + sin\alpha) + \frac{f_{h,1}\ s_1 d}{1+\beta}\ (1 - \mu\ tan\alpha)\left[ \sqrt{\beta + 2\beta^2\left(1 + \frac{s_2}{s_1} + \left(\frac{s_2}{s_1}\right)^2 + \beta^3\left(\frac{s_2}{s_1}\right)^2\right)} - \beta\left(1 + \frac{s_2}{s_1}\right) \right]$$

$$(d)\ R_d = R_{ax}\ (\mu\ cos\alpha + sin\alpha) + \frac{f_{h,1}\ s_1 d}{2+\beta}\ (1 - \mu\ tan\alpha)\left[ \sqrt{2\ \beta\ (1+\beta) + \left(\frac{4\ \beta\ (2+\beta)M_y}{f_{h,1}\ d\ s_1^2}\right)} - \beta \right]$$

$$(e)\ R_e = R_{ax}\ (\mu\ cos\alpha + sin\alpha) + \frac{f_{h,1}\ s_1 d}{1+2\beta}\ (1 - \mu\ tan\alpha)\left[ \sqrt{2\ \beta^2\ (1+\beta) + \left(\frac{4\ \beta\ (1+2\beta)M_y}{f_{h,1}\ d\ s_1^2}\right)} - \beta \right]$$

$$(f)\ R_f = R_{ax}\ (\mu\ cos\alpha + sin\alpha) + (1 - \mu\ tan\alpha)\sqrt{\frac{2\beta}{1+\beta}}\left[ \sqrt{2\ M_y\ f_{h,1}\ d\ cos^2\alpha} \right]$$

where:

- $F_{v,R}$ is the load carrying capacity of timber-to-timber connection with inclined screw.
- $R_{ax}$ is the withdrawal capacity of the screw.

- $s_1$ is the anchorage length of the screw into the first wood element measured orthogonally to the shear plane.
- $s_2$ is the anchorage length of the screw into the second wood element measured orthogonally to the shear plane.
- $f_{h,1}$ is the embedment strength of the head side wood member.
- $f_{h,2}$ is the embedment strength of the point side wood member.
- $\beta = \frac{f_{h,2}}{f_{h,1}}$ the ratio of embedment strengths.
- $R_{ax,1}$ is the withdrawal strength of the screw from the head side wood member.
- $R_{ax,2}$ is the withdrawal strength of the screw from the point side wood member.
- $d$ is the effective diameter of the screw.
- $M_y$ is the yield moment of the screw.
- $\mu$ is the friction coefficient at the interface between wood elements (0.25 for wood to wood).

The characteristics of Connection-B are such that two separate screws and three timber members act together to provide load resistance. The first screw is 150 mm long and connects the 76 mm wide LSL wedge piece (side member) with the 38 mm wide LSL stud member (main member), whereas the second screw was 200 mm long and attached the LSL wedge piece (side member) to the 76 mm thick top plate of the support wall (Figure 27). The two screws have a diagonal spacing of 95 mm (which is greater than the 12 times fastener diameter) in the side member wedge piece (Figure 27).

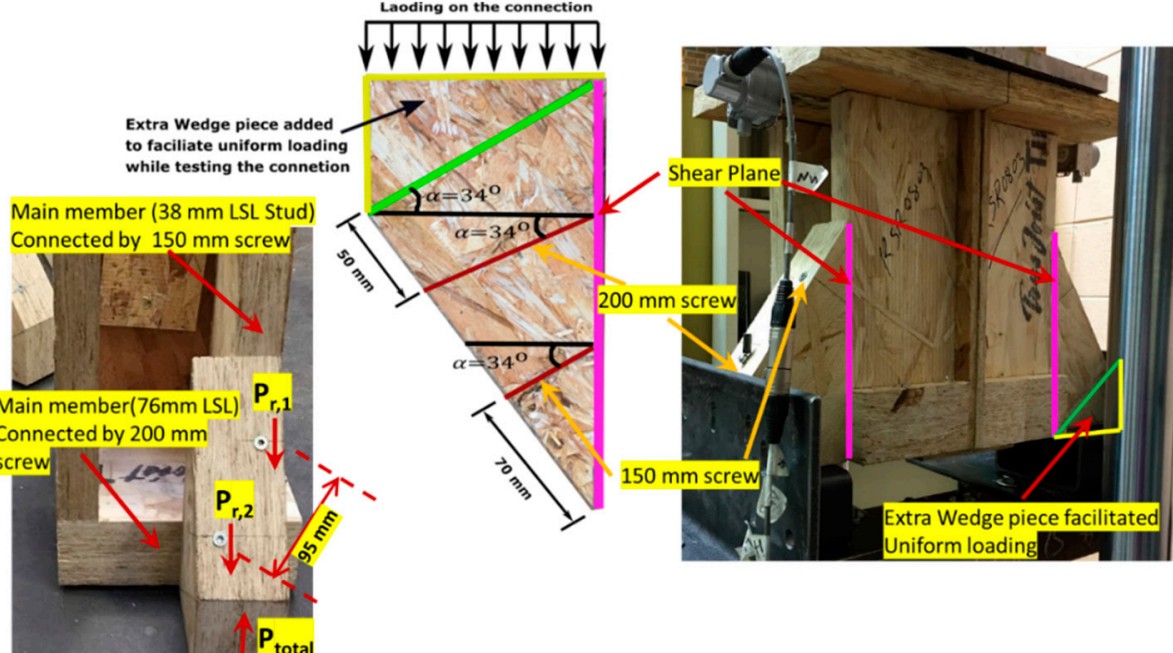

**Figure 27.** Support wall-to-wedge piece connection details for strength prediction.

When the connection is loaded, each screw contributes simultaneously to resist the applied force. Therefore, the theory proposed by Bejttka and Blaß [28] was applied, adopting the following assumption: (a) the total load carrying capacity of the joint will be the combined ultimate loads of the fasteners ($P_{total} = P_{r,1} + P_{r,2}$) as illustrated in Figure 27; (b) the system parameter embedment strength of timber ($f_h$) depends on properties such as screw geometry, surface roughness, or load to the grain direction of timber. Therefore, the parallel to grain direction, in this case, is considered along the strand direction of LSL (Figure 28a). It should be noted that the side member embedment strength is parallel to the strand direction while the main members are at an inclination equal to ($90°$-roof angle).

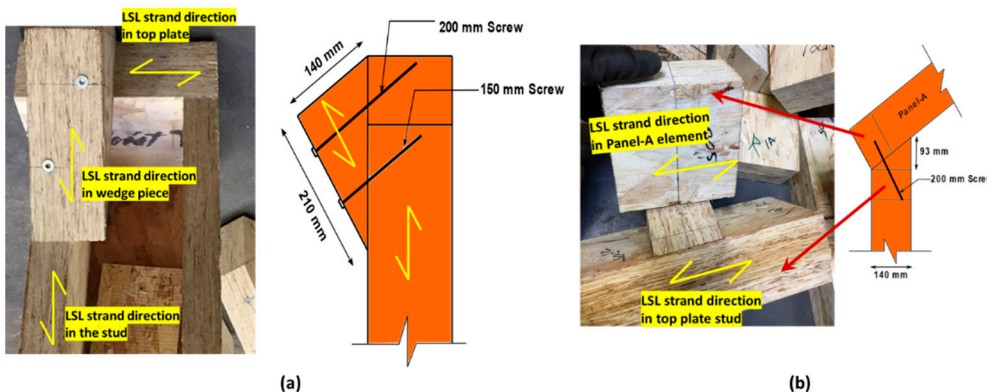

**Figure 28.** Strand direction in: (**a**) Connection-B; (**b**) Connection-A.

The application of the above model requires estimation of embedment strength ($f_h$) and screw withdrawal resistance ($R_{ax}$) of corresponding connection members. The embedment strength of connection members was predicted using the recent Canadian study by Khan et al. [29] for various inclinations of the screw axis with respect to the grain as follows:

$$f_{h,\alpha} = \frac{0.206 \cdot \rho_k^{0.860} \cdot d^{-0.0078}}{2.89 \cdot cos^2\alpha + sin^2\alpha} \tag{4}$$

For the HBS screw, $R_{ax1}$ of the side member was assumed to equal the minimum value between the head pull-through resistance ($R_{head}$) and the tensile strength of the screw. In the withdrawal test of the HBS screw, head pull-through resistance was observed to be the failure mode. Therefore, $R_{ax1}$ is equal to $R_{head}$ for the HBS screw. For SDWS screw, $R_{ax1}$ of the side member is the tensile strength since the withdrawal test failure was the fracture of the threaded part of the screw. The SDWS screw head diameter is large (19 mm), and the geometry prevents head pull out of the side members. In the case of the main member, $R_{ax2}$ is the axial resistance of the screw corresponding to the lower value of the thread withdrawal resistance ($R_{thread}$) and the tensile strength of the screw ($R_{tens}$). The reported value of $R_{thread}$ is governed by both screws. With regards to Equation (3), $R_{ax}$ is the minimum of $R_{ax1}$ and $R_{ax2}$. Therefore, $R_{ax} = R_{head}$ for the HBS screw and $R_{ax} = R_{thread}$ for the SDWS screw were used in predicting connection capacity. Therefore, in applying the above model, screw withdrawal capacity ($R_{ax}$) obtained from the Connection-A withdrawal test was used.

In the case of Connection-A, the wedge piece was glued on the top plate part, and therefore, it can be assumed that in connection configuration, they act as a single member (in this case, it is regarded as the side member). The screw was inserted at an angle of 60° with the strand direction of the side member to connect it to the main member that represents the panel-A element (Figure 28b), and thus, it was inclined at 4° with the line perpendicular to the shear plane (Figure 29). For such a small angle, the effect of screw inclination can be ignored since the connection capacity increases with the angle between 15° and 50° [28]. Therefore, the analytical validation was performed according to expressions reported in Section 8.7.2 of Eurocode 5 [20] and CSA O86-19 [7] expressions reported in Section 12.6.

Table 4 summarizes the calculated connection capacities and the failure modes using the above-mentioned models, and Figure 30 illustrates the comparison of predicted capacity with experimental mean maximum load values. The model predicted the same failure mode as the test for Connection-B. Overall, it can be concluded that the proposed model by Bejttka and Blaß [28] predicts reasonable strength of Connection-B (Figure 30). Therefore, this model can be used to obtain the design value of this connection for other roof slopes. The predicted capacity of Connection-A using Eurocode 5 [20] and CSA 086-19 [7] is close to the tested value (Figure 30), and the failure mode matched the observed values in the opened tested specimens. Therefore, it is evident from the experimental results that available analytical models can be used to obtain reasonable estimates of design values for

the connections of the novel panelized roof. However, there were uncertainties associated with some material input properties, such as embedment strength. Consequently, further studies are highly recommended in order to improve the calibration of material input properties for the theoretical models.

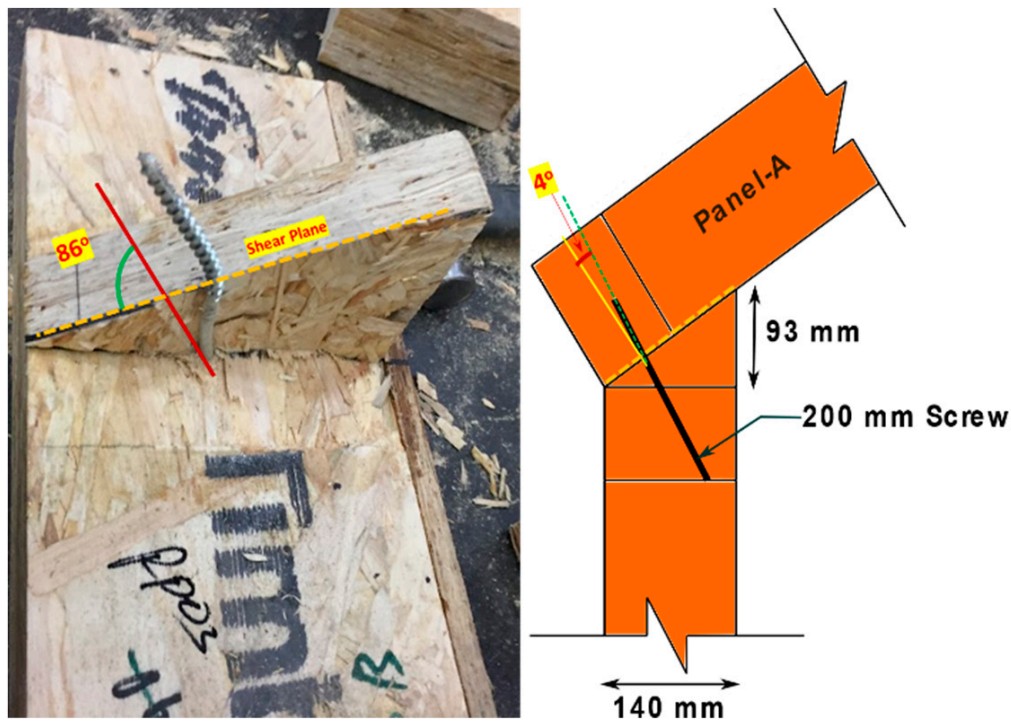

**Figure 29.** Connection -A shear plane.

**Table 4.** Theoretical load-bearing capacity calculation.

| | Connection-B | | Connection-E | Connection-A | | | |
|---|---|---|---|---|---|---|---|
| | HBS Screw | SDWS Screw | HBS Screw | HBS Screw | | SDWS Screw | |
| | Bejttka and Blaß Model | | | Eurocode | CSA 086 | Eurocode | CSA 086 |
| $R_{ax}$ (kN) | 5.88 | 9.20 | 5.88 | 5.88 | - | 9.20 | - |
| $f_{h,1,k}$ (N/mm$^2$) | 12.95 | 12.70 | 12.95 | 25.42 | 25.42 | 24.93 | 24.93 |
| $f_{h,2,k}$ (N/mm$^2$) | 23.52 | 23.07 | 23.52 | 12.99 | 12.99 | 12.74 | 12.74 |
| $M_{y,k}$ (Nmm) | 9494 | 25,590 | 9494 | 9494 | 9494 | 25,590 | 25,590 |
| $d$ (mm) | 6.00 | 7.70 | 6.00 | 6.00 | 6.00 | 7.70 | 7.70 |
| $F_{max,\,model}$ (kN) | 10.77 | 17.10 | 5.46 | 3.08 | 2.91 | 5.26 | 4.9 |
| Mean $F_{max,\,experiment}$ (kN) | 12.84 | 21.27 | 5.77 | 4.53 | 4.53 | 5.70 | 5.70 |
| Safety factor, $\eta = \frac{F_{max,\,experiment}}{F_{max,\,model}}$ | 1.19 | 1.24 | 1.06 | 1.47 | 1.56 | 1.08 | 1.17 |
| Failure mode | Two plastic hinges per share plane | Two plastic hinges per share plane for 200 mm screw; one plastic hinge for 150 mm screw | Two plastic hinges per share plane | Two plastic hinges per share plane in screw | | | |

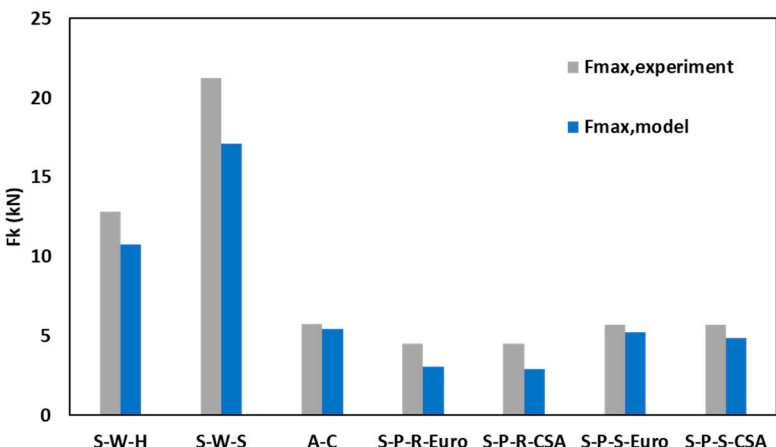

**Figure 30.** Comparison between the experimental and theoretical results in terms of capacity *. *
Note: S-W-R = Connection -B with HBS screw, S-W-S = Connection -B with SDWS, A-C = Apex
connection, S-P-R-Euro = Connection -A with HBS screw prediction using Eurocode 5, S-P-S-Euro =
Connection -A with SDWS screw prediction using Eurocode 5, S-P-R-CSA = Connection -A with HBS
screw prediction using CSA 086-19, S-P-S-CSA = Connection -A with SDWS screw prediction using
CSA 086-19.

## 9. Conclusions

The connections of a novel panelized roof system have been designed and developed.
The main goal of this study was to obtain the strengths of proposed connections applicable
to roof manufacturing in offsite production facilities. Based on testing of the seven con-
nection configurations, the load-bearing capacity, the load-slip behaviour, and the failure
modes were investigated and described. These test data can be used to derive design
properties for structural design purposes.

Furthermore, an assessment of maximum wind uplift capacity from the test of common
connections used on North American roofs shows the proposed connections have adequate
capacity and even, in some cases, are higher than those commonly used in light-frame wood
construction. For instance, the average wind uplift capacity of the Connection-A connection
capacity varied between 5.88 and 9.2 kN depending on the screw diameter, whereas the
tested capacity of the hurricane tie steel plate connection—a very common mechanism
used to connect the truss bottom chord with light-frame shear walls in North America,
has an average maximum capacity of 3.9 to 5.9 kN as reported in the test conducted
by Alhawamdeh and Shao [30] and Canino et al. [31]. Thus, this study demonstrates
self-tapping screws are suitable for connection design in panelized roof fabrication with
engineered wood products such as LSL.

The limitation of analytical model validation was the lack of material test data. No
embedment test of LSL was performed. Embedment strength depends on the screw
diameter and density of the wood. The embedment strength equation used here is primarily
developed for wood products, and the LSL density was regarded as the equivalent of
Douglas Fir wood species. Consequently, further validation of embedment strength is
required before implementing the analytical models. Another limitation was the lack
of testing of screw properties. As can be observed from Table 1, SDWS screws have a
higher yield moment capacity than HBS screws but lower tensile strength. Therefore,
future studies should include embedment strength equation development for LSL and
experimental evaluation of fastener yield strength in bending.

**Author Contributions:** Conceptualization, M.S.I.; data curation, M.S.I.; formal analysis, M.S.I.; fund-
ing acquisition, Y.H.C.; investigation, M.S.I.; methodology, M.S.I. and Y.H.C.; project administration,
M.S.A.; resources, Y.H.C. and M.S.A.; supervision, Y.H.C.; validation, M.S.I. and Y.H.C.; visualization,
M.S.I.; writing—original draft, M.S.I.; writing—review and editing, Y.H.C. All authors have read and
agreed to the published version of the manuscript.

**Funding:** This work was supported by a grant from the Natural Sciences and Engineering Research Council (NSERC) of Canada through the Engage Grant and Industrial Research Chair programs.

**Institutional Review Board Statement:** Not applicable.

**Informed Consent Statement:** Not applicable.

**Data Availability Statement:** The data presented in this study are available upon request from the corresponding author.

**Acknowledgments:** We thank ACQBUILT Inc., Rothoblaas, and Simpson Strong-Tie Inc. for material supplies and technical support.

**Conflicts of Interest:** The authors declare no conflict of interest.

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
