# Peer review of "Design and Experimental Analysis of Connections for a Panelized Wood Frame Roof System"

_buildings, doi:10.3390/buildings12060847_

Round 1
Reviewer 1 Report
The manuscript is easy to read and well organized. The topic fulfill the aims and scope of the journal. The research is presented in detail and the information presented is relevant for practitioners.
Nonetheless, I have some comments that can be found in the attachment.

Reviewer 2 Report
The authors conducted mechanical tests with various configurations which are used for a novel panelized roof system. Although an academic level is not so high, this study may contribute to promote the productive building systems. There are a few points which should be modified before publishing.
Table 2
The unit of MOE should be added.
Figures 8-11
It is difficult for the readers to understand the photo. I recommended that the authors add symbol on it as shown in attached image.
Line 310
I can’t understand why the authors used average value in maximum force. In general, the individual specimens’ value of maximum force is used for calculating the stiffness. Is the method author describe is also general?
Lines 315 and 322
The authors used the terminologies “maximum force” and “peak load”. If their meaning is same, the authors should use the same terminology throughout the paper.

Reviewer 3 Report
Dear authors,
thank you very much for the interesting research. The article is within the scope of the journal.
The manuscript deals with experimental and analytical analyses of two types of steel screws in a panelized roof. The roof panels are made of LSL or OSB. Experimental results are comprehensive and clearly explained. The paper is interesting to timber engineering society.
As the paper is clearly written and conclusions are supported by results, I just have three remarks.
1. Can you be more specific in the title of the manuscript? The title is too broad and the words "novel" and "connection" will not attract new readers. Please specify it more clearly
2. line 232: why 2.54 mm/min? How did you calculate this value?
3. STAR documentation could be better. You have 26 references but just 9 of them are scientific articles. The rest is either a standard or your previous papers. A lot of research on the topic was carried out in Europe.
